# Multiple Sclerosis CD49d^+^CD154^+^ As Myelin-Specific Lymphocytes Induced During Remyelination

**DOI:** 10.3390/cells9010015

**Published:** 2019-12-19

**Authors:** Paweł Piatek, Magdalena Namiecinska, Małgorzata Domowicz, Marek Wieczorek, Sylwia Michlewska, Mariola Matysiak, Natalia Lewkowicz, Maciej Tarkowski, Przemysław Lewkowicz

**Affiliations:** 1Department of Neurology, Laboratory of Neuroimmunology, Medical University of Lodz, Poland, ul. Pomorska 251, 92-213 Lodz, Poland; pawel.piatek@umed.lodz.pl (P.P.); magdalena.namiecinska@umed.lodz.pl (M.N.); malgorzata.domowicz@umed.lodz.pl (M.D.); mariola.swiderek-matysiak@umed.lodz.pl (M.M.); 2Department of Neurobiology, Faculty of Biology and Environmental Protection, University of Lodz, 90-236 Lodz, Poland; marek.wieczorek@biol.uni.lodz.pl; 3Laboratory of Microscopic Imaging and Specialized Biological Techniques, Faculty of Biology and Environmental Protection, University of Lodz, 90-237 Lodz, Poland; sylwia.michlewska@biol.uni.lodz.pl; 4Department of General Dentistry, Medical University of Lodz, 92-213 Lodz, Poland; natalia.lewkowicz@umed.lodz.pl; 5Department of Biomedical and Clinical Sciences, Luigi Sacco, University Hospital, University of Milan, 20122 Milano, Italy; maciej.tarkowski@outlook.it

**Keywords:** multiple sclerosis, myelin-specific lymphocytes, remyelination

## Abstract

Multiple sclerosis (MS) is a demyelinating autoimmune disease of the central nervous system (CNS) mediated by autoreactive lymphocytes. The role of autoreactive lymphocytes in the CNS demyelination is well described, whereas very little is known about their role in remyelination during MS remission. In this study, we identified a new subpopulation of myelin-specific CD49d^+^CD154^+^ lymphocytes presented in the peripheral blood of MS patients during remission, that proliferated in vitro in response to myelin peptides. These lymphocytes possessed the unique ability to migrate towards maturing oligodendrocyte precursor cells (OPCs) and synthetize proinflammatory chemokines/cytokines. The co-culture of maturing OPCs with myelin-specific CD49d^+^CD154^+^ lymphocytes was characterized by the increase in proinflammatory chemokine/cytokine secretion that was not only a result of their cumulative effect of what OPCs and CD49d^+^CD154^+^ lymphocytes produced alone. Moreover, maturing OPCs exposed to exogenous myelin peptides managed to induce CD40-CD154-dependent CD49d^+^CD154^+^ lymphocyte proliferation. We confirmed, in vivo, the presence of CD49d^+^CD154^+^ cells close to maturating OPCs and remyelinating plaque during disease remission in the MS mouse model (C57Bl/6 mice immunized with MOG_35-55_) by immunohistochemistry. Three weeks after an acute phase of experimental autoimmune encephalomyelitis, CD49d^+^/CD154^+^ cells were found to be co-localized with O4^+^ cells (oligodendrocyte progenitors) in the areas of remyelination identified by myelin basic protein (MBP) labelling. These data suggested that myelin-specific CD49d^+^CD154^+^ lymphocytes present in the brain can interfere with remyelination mediated by oligodendrocytes probably as a result of establishing proinflammatory environment.

## 1. Introduction

Multiple sclerosis (MS) is an autoimmune lymphocyte-dependent demyelinating disease of the central nervous system (CNS). Axonal loss and functional disability in forming myelin sheaths by oligodendrocytes (OLs) are two important processes responsible for the disease progression and patient death within years [1]. Relapsing-remitting MS (RR-MS) is the most prevalent form affecting mainly young people, which is characterized by acute attacks (relapses) followed by a period of partial withdrawal of symptoms (remission) [2]. Relapse mainly involves the brain areas previously affected by the disease where gradual, extensive processes of demyelination and irreversible neuron impairment had already taken place [3]. Slow but constant progression of the disease is caused by inefficient and only partial regeneration of the myelin sheath around the neuronal axons after primary attack or relapse [4]; what still remains is the unsolved aspect of MS pathology. This might be a result of incomplete clearance of inflammatory cells from the brain during disease remission or proliferation of autoreactive cells within remyelinating plaques. Alternatively, autoreactive myelin-specific lymphocytes might migrate from the periphery to the CNS, preferably to the areas of previous demyelination [5].

Indeed, presence of inflammatory cells was revealed in MS lesions regardless of the clinical diagnosis [3]. CD154 expression was previously demonstrated to be highly specific for human and mouse antigen-specific lymphocytes [6], which was found in MS brain, but not in the healthy CNS, or in the course other neurodegenerative diseases [7]. In a mouse model of RR-MS, experimental autoimmune encephalomyelitis (EAE), it was demonstrated that CD154^+^ T cells infiltrate the CNS on the fourth day of postimmunization and their number increases during the acute phase and remains constant in the period of remission [8]. CD154-mediated activation of CD40-expressing cells in the CNS such as monocytes, macrophages, and activated microglia [7] results in the secretion of proinflammatory cytokines/chemokines, which fuel ongoing inflammation [9]. The significance of CD154-CD40 dyad in the disease progression has been demonstrated in the studies with CD154 and CD40 knock-out mice, as well as with the use of neutralizing mAbs anti-CD154 or anti-CD40 in EAE [7,10,11,12,13,14]. The role of CD154-CD40 dyad was also confirmed in human studies. The single nucleotide polymorphism (SNP) analysis revealed that the variant of CD40 gene (rs1883822C>T) was associated with an increased risk for MS in comparison to healthy individuals about 1.5-fold in heterozygous and 2.5-fold in homozygotes, respectively [15,16]. It was also demonstrated that stimulation of CD40 receptor on B cells of RR-MS patients resulted in significantly higher proliferation than in healthy subjects [17], and application of neutralizing anti-CD154 mAbs (clone IDEC-131) showed promising results in inhibiting the relapses during clinical trials [9,18].

To initiate CNS inflammation, myelin-specific T cells must be first activated in the periphery, gain access to the CNS, and then be reactivated by a self-antigen [5]. CD49d is a major integrin that allows autoreactive cells to cross the blood–brain barrier (BBB). It was previously revealed that CD49d are expressed on lymphocytes of MS patients at significantly higher expression levels than in healthy people [19]. The efficiency of neutralizing anti-CD49d (natalizumab) was already demonstrated in clinical trials in MS [18]. It seems that the ability of autoreactive lymphocytes to migrate towards remyelinating plaque can be dependent on the chemotactic gradient created by resident brain cells or residual brain-infiltrating immune cells.

Based on these data, we assumed that MS autoreactive lymphocytes induced in the periphery must express both CD49d and CD154 receptors. Further, oligodendroglia precursor cells (OPCs), as a natural reservoir for oligodendrocytes during remyelination, might be the source of chemoattractants for peripheral autoreactive lymphocytes enabling directed migration to the remeylinating plaques.

In this study, we have characterized myelin-specific CD49d^+^CD154^+^ lymphocytes in the peripheral blood of RR-MS patients and demonstrated the potential mechanisms that facilitate interaction of these cells with oligodendrocytes and augment proinflammatory reaction. Our findings can give a better understanding of mechanisms that lead to the inhibition of remyelination and MS progression.

## 2. Materials and Methods

### 2.1. Human Subjects

All participants of the study were diagnosed and recruited at the Department of Neurology, Medical University of Lodz. Ten patients (6 females, 4 males; average age 43.2 ± 7.23) diagnosed with RR-MS according to the McDonald criteria 2010 [20] were enrolled into the study during remission. Mean disease duration was 5.4 ± 1.49 years and mean time from last relapse was 8 months. Relapse was defined as the appearance of new neurological signs or worsening of pre-existing ones after a minimum 30 days of clinical stability. Remission was defined as a minimum 3 months of neurological stabilization after last relapse. None of the patients received systemic steroids or other anti-inflammatory or immunosuppressive drugs for at least three months prior to the study. For the control group, 10 healthy volunteers matched by age and sex were recruited. Then, 20 mL of peripheral blood was collected from the patients during remission phase of the disease and healthy controls (HC). The study was approved by the Ethics Committee of the Medical University of Lodz (RNN/44/14/KE and RNN/28/18/KE), and informed consent was obtained from each participant of the study.

### 2.2. EAE Mouse Model

For the mouse model of MS, C57Bl/6 mice were housed and maintained in an accredited facility, the Animal Core Department of the Medical University of Lodz. Six eight-week-old female C57Bl/6 mice were injected subcutaneously with MOG_35-55_ peptide (NH2-MEVGWYRSPFSRVVHLYRNGK-COOH) in complete Freund adjuvant (CFA). On day 0, each mouse received 0.25 mL of 0.15 mg mixture of dissolved MOG_35-55_ peptide in 0.1 mL of PBS and 0.75 mg of Mycobacterium tuberculosis in 0.15 mL of CFA, injected in four abdominal sites. The BBB permeability was increased by using 0.2 μg Pertussis toxin (Sigma) injected into a tail vein on days 0 and 2. Mice were observed for neurologic signs of EAE and were scored using the scale 0–5 as follows: 0—non disease; 1—weak tail or wobbly walk; 2—hind limb paralysis; 3—forelimbs paralysis; 4—hind and forelimb paralysis; 5—death or euthanasia for ethical reasons. The neurological signs of the EAE mice at the disease peak were scored for 4 and reduced to 1 three weeks after the disease peak (remyelination period). Thus, mice were sacrificed and sampled to further analysis three weeks after the disease peak. For the control group, four healthy female C57Bl/6 mice, selected by weight (average 15.7 ± 1.21 g), were chosen. All experiments were approved by the University Ethics Committee (12/ŁB702/2014) and the animal study was arranged in accordance with ARRIVE guidelines [21].

### 2.3. Human Cell Model of Progenitor Cells Differentiation to Mature Myelin Producing Oligodendrocytes

A human oligodendroglia cell line MO3.13 (OLs, Tebu-bio, Le Perray En Yvelines, France) was used as the model of progenitor cell differentiation to mature myelin-producing OLs. We chose the MO3.13 cell line, which differentiates after phorbol 12-myristate 13-acetate (PMA) stimulation, as the most adequate model of OPC maturation. Opposite to other OL lines (HOG or KG-1C), MO3.13 cells during maturation exhibited the strongest similarity to primary human OLs in morphology as well as in gene and protein expression [22]. OLs were cultured in DMEM-high glucose medium supplemented with 10% fetal bovine serum, 1% Penicillin-Streptomycin, and maintained at 37 °C with 5% CO_2_ in humidified atmosphere. The cultures were conducted in 75 cm^2^ flasks (Nunc, Thermo Scientific, Rostilde, Denmark). After 80% confluence, cells were passaged by using a 0.25% trypsin-EDTA solution in total three times for a week with a dilution factor of 1/8. To induce OPC maturation, Phorbol 12-Myristate 13-Acetate was added (0.1 mM) for 72 h and cells were incubated at 37 °C with 5% CO_2_ in humidified atmosphere. All reagents used for culturing were purchased from Sigma-Aldrich.

### 2.4. RR-MS CD49d^+^CD154^+^ Lymphocyte Sorting

Peripheral blood mononuclear cells (PBMCs) were isolated by density centrifugation over Lymphoprep (Axis-Shield, Oslo, Norway) according to manual instruction. PBMCs were stimulated by adding a mixture of peptides: Proteolipid protein (PLP_139-151_ NH2-HSLGKWLGHPDKF-COOH; pepPLP), myelin oligodendrocyte glycoprotein (MOG_35-55_ (NH2-MEVGWYRSPFSRVVHLYRNGK-COOH; pepMOG), and myelin basic protein (MBP_85-99_ NH2-VHFFKNIVTPRTPPP-COOH; pepMBP) (each 25 μg/mL) for 21 h (37 °C, 5% CO_2_ in humidified atmosphere). Following the incubation, cells were washed and labelled with 5 μg/mL CD154-PE (clone 89-76, BD Biosciences, San Jose, California, USA) and CD49d-FITC (clone 9F10, BD Biosciences) mAbs for 30 min at 4 °C. CD49d^+^CD154^+^ population was isolated by FACS sorting procedure using a FACSAria with purity mask mode (BD Biosciences). The purity of FACS-sorted CD49d^+^CD154^+^ cell fraction assessed by two-color flow cytometry was ~99.5% (Figure 1D).

### 2.5. hOPC-Lymphocyte Co-Culture

To examine the effect of interaction of myelin-specific CD49d^+^CD154^+^ lymphocytes with maturating OPCs on the environment, 4 × 10^4^ lymphocytes were added to 2 × 10^6^ OPCs (1:50) seeded on 6-well plates 24 h earlier. PMA (phorbol 12-myristate 13-acetate), artificial stimulator of OPC differentiation, was added (0.1 mM) to culture in DMEM-high glucose medium supplemented with 5% FBS, 1% penicillin-streptomycin at 37 °C with 5% CO_2_ for 72 h.

### 2.6. Analysis of RR-MS Myelin-Specific CD49d^+^CD154^+^ Cell Proliferation using CFSE Method

Myelin-specific CD49d^+^CD154^+^ cell proliferation was analyzed with the use of 5,6-carboxyfluorescein diacetate succinimidyl ester (CFSE, BD Biosciences) [23]. CFSE-labelled PBMCs were incubated in 6-well culture plates at the density of 2 × 10^6^ cells per well with or without pep(MOG/MBP/PLP) or pre-incubated with pep(MOG/MBP/PLP) (21 h, each 25 μg/mL) maturating OPCs (1:50). For neutralizing experiments, 5 μg/mL anti-hCD40 (clone 40804) or isotype control (clone 20102R, all R&D Systems) were added 10 min before pep(MOG/PLP/MBP) stimulations or 10 min to pre-incubated with pep(MOG/PLP/MBP) hOPCs and washed in RPMI. After 24, 48, and 72 h of incubation, cells were labelled with CD49d (clone 9F10, BD Biosciences), CD154 (clone 89-76, BD Biosciences), or mouse IgG1 isotype control (clone 11711, R&D system) antibodies and analyzed by BD LSR Flow Cytometer.

### 2.7. Flow Cytometry Analysis of CXCR4, CXCR7, CCR6 Surface Expression on CD49d^+^CD154^+^ Lymphocytes, and CD40 on OPCs

First, 6 × 10^5^ of sorted CD49d^+^CD154^+^ lymphocytes stimulated or not with MOG/PLP/MBP peptides were labelled with monoclonal antibodies anti-CXCR4 APC (clone 205410), anti-CXCR7 APC (clone 150503), anti-CCR6 APC (clone 53103), or mouse IgG2B APC isotype control (clone 133303, all from R&D Systems). Anti-human CD40 PECy5 (clone 5C3, BD Biosciences) and mouse IgG1κ as isotype control (clone MOPC-21, BD Biosciences) were used to analyze CD40 expression changes during OPC maturation. After a 30-min incubation at RT, all samples were rinsed, fixed by 1% formaldehyde solution, and analyzed (LSRII, BD).

### 2.8. Chemotaxis Assay 

Chemotaxis was determined by Boyden chamber cell-insert. As a chemotactic gradient, we used supernatants derived from the culture of maturing OPCs. Isolated myelin-specific CD49d^+^CD154^+^ and CD49d^−^CD154^−^ lymphocytes obtained from RR-MS patients were incubated in the upper chamber of cell culture insert (3.0 μm φ, ThinCert, Greiner Bio-One, Solingen, Germany) immersed in the OPC culture supernatant (low chamber) for 2 h at 37 °C. Lymphocytes that migrated through the cell culture insert were labeled with CD3(FITC)/CD19(PE)/CD45(PerCP) (clone SK7/SJ25C1/2D1, BD Biosciences), analyzed by flow cytometry (LSR II, BD), and normalized using tubes containing constant number of microbeads (Trucount™, BD Biosciences). As the negative control, we counted the number of lymphocytes in the lower and upper chambers using PBS instead of supernatant.

### 2.9. Human Chemokine Multiple Profiling Assays

Chemokine and cytokine concentrations in co-culture supernatants of maturing OPCs with CD49d^+^CD154^+^ lymphocytes as well as RR-MS and HC PBMC with or without MOG/PLP/MBP peptides were measured using Bio-Plex Pro™ Human Chemokine Assays (Bio-Rad Laboratories, Hercules, California, USA). Standards and samples were diluted (1:4) in sample diluent and transferred to the plate containing magnetic beads for 1 h at RT. Next, the plate was washed (3×) and detection antibody was added for 30 min on a shaker (850 rpm) at RT. After that, the plate was washed (3×) and streptavidin-PE solution was added for 10 min. Subsequently, the plate was washed (3×) and samples were re-suspended in 125 µL of assay buffer and analyzed within 15 min. All samples were analyzed at the same time in duplicates. All reagents and technology were provided by Bio-Rad Laboratories (Bio-Plex 200).

### 2.10. Mouse CNS Histopathological Examination

Three weeks after peak of the disease, representative animals were perfused with 25 mL of cold-buffered 2.5% glutaraldehyde and ice-cold PBS (10 mL/min). Brains and spinal cords were removed, fixed in 4% formaldehyde solution, and hematoxylin-eosin 8 μm thick slices were examined for CNS pathology (all reagent form Sigma Aldrich). The histopathological visualization of myelin and neuron arrangement was performed using BrainStain™ Imagining Kit (Molecular Probes™, Invitrogen, Paisley, UK) according to the manual instruction. The kit contained FlouroMyelin™Green, which selectively stains myelin; NeuroTrace^®^ 530/615, which selectively stains neuron cell bodies; and DAPI, which binds to DNA.

### 2.11. Immunohistochemistry (IHC)

For IHC analysis, brains were cut into 12 μm sections at −20 °C using the cryostat microtome method (Leica CM1950, Nussloch, Germany) and transferred to gelatin-coated microscope slides. Samples were fixed with ice-cold acetone (10 min) and washed in PBS. Subsequently, slides were blocked (10% goat blocking serum in PBS with 0.3% Triton-X 100 and 0.01% sodium azide; 20 min, RT) and washed. Samples were incubated overnight with first antibody mix containing: O4, (clone O4, 5 μg/mL, R&D Systems), MBP (clone F-6, 1:100, Santa Cruz Biotechnology, Dallas, TX, USA), CD49d (clone PS/2, 5 μg/mL, Abcam), and CD154 (clone JM11-34, 10 μg/mL, ThermoFisher Scientific). Secondary Abs: Goat anti-mouse AlexaFluor 350 (ThermoFisher Scientific 5 μg/mL), goat anti-rat TRITC (Sigma-Aldrich 1:100), and goat anti-rabbit FITC (Invitrogen 1:100) were applied for 1 h at RT, and rinsed several times. Images were acquired using a confocal microscope Leica TCS LSI confocal microscope with 10 and 63 × objective. The image of the whole brain was prepared from a montage of 25 single pictures. Fluorescence intensity was determined as the average fluorescence (Avg. area), the sum of the fluorescence from all segments divided by the number of segments (IPLab Pathway 4.0 software, BD Bioscience). The average fluorescence for each fluorochrom was calculated using 31 single regions (region of interest, ROI) taken from four independent experiments (one slide per experiments). The level of baseline fluorescence was established individually for each experiment. Nonspecific fluorescence (signal noise) was electronically diminished to the level when nonspecific signal was undetectable [24]. IHC data were additionally presented as the values of overlap coefficient that indicates the overlap of the fluorescence signals between the channels FITC, AlexaFluor 350, and TRITC. It was calculated as the mean value from every single ROI using Leica Microsystem (LAS - X, ver. 3.7.020979 software, Leica, Germany). The overlap coefficient ranges from 0 (no co-localization) to 1 (complete co-localization).

### 2.12. Statistics

Arithmetic means and standard deviations were calculated for all parameters from at least four independent experiments. A statistical analysis of differences was performed using the one-way ANOVA test. Tukey’s test was used for multiple comparisons as a post-hoc test when statistical significances were identified in the ANOVA test. Statistical significance was set at *p* < 0.05.

## 3. Results

### 3.1. CD49d^+^CD154^+^ Lymphocytes of RR-MS Patients Proliferate in Vitro in Response to Myelin Proteins

First, using flow cytometry, we analyzed the rate of CD49d^+^CD154^+^ lymphocytes in PBMCs isolated from RR-MS patients during the disease remission or in healthy controls (HCs). PBMCs were incubated with or without myelin peptides MOG/PLP/MBP for 72 h. We found that RR-MS PBMCs contained more CD49d^+^CD154^+^ lymphocytes in comparison to HC PBMCs (1.2% vs. 0.4%), and the number of CD49d^+^CD154^+^ lymphocytes was further increased (2.8–3.3%) after the incubations of RR-MS PBMCs with myelin peptides (Figure 1A). To determine whether myelin peptides induce CD49d^+^CD154^+^ cell proliferation, we additionally performed CFSE analysis. We found that contrary to HC, CD49d^+^CD154^+^ lymphocytes from RR-MS patients proliferated in response to myelin peptides (1% before stimulation vs. 73% after myelin stimulation), and only 2% of CD49d^+^CD154^+^ lymphocytes remained in the region of non-proliferating cells (CFSE^high^) (Figure 1B). CD40 and its ligand CD154 are key players in T cell-B cell and T cell-antigen-presenting cell (APC) interactions [25]. We demonstrated that proliferation of RR-MS CD49d^+^CD154^+^ lymphocytes in response to myelin peptides was dependent on CD154-CD40 interaction as it was partially abrogated by CD40 neutralizing antibodies (Figure 1B). Additionally, CD49d^+^CD154^+^ lymphocyte proliferation was accompanied by enhanced concentration of sCD40 in supernatants as a result of negative feedback regulatory loop for CD154-CD40 interaction (Figure 1C) [26]. RR-MS PBMCs in response to myelin peptides opposite to HC PBMCs produced proinflammatory chemokines: CCL1, 2, 7, 8, 20, 21, 22, 24, 25, 27, CXCL1, 2, 5, 6, 8, 9, 11, 17, proinflammatory cytokines: MIF1, IL-1β, IL-2, and TNF-α (Table 1). Taken together, we demonstrated the induction of autoreactive CD49d^+^CD154^+^ lymphocytes in MS patients in PBMCs exposed to myelin peptides and confirmed the role of CD154-CD40 interaction in this process. Therefore, we applied high-purity FACS sorting to isolate CD49d^+^CD154^+^ lymphocytes from RR-MS patients and HCs for further experiments (Figure 1D).

### 3.2. Interaction of RR-MS CD49d^+^CD154^+^ Lymphocytes with Maturing Human OPCs (hOPCs) Generates Positive Proinflammatory Feedback Loop

Human MO3.13 cells stimulated by PMA were used as the cellular model of OPC polarization to mature OLs (Figure 2A). We first demonstrated that human OPCs (hOPCs) during maturation synthetized factors capable of leucocyte recruitment and polarization: CCL2, CCL20, CXCL1, CXCL2, CXCL5, IL-8 (CXCL8), CXCL9, CX3CL1, MIF1, and IL-6 (Figure 2B, left panel, Table 2).

To evaluate the capability of RR-MS CD49d^+^CD154^+^ lymphocytes to migrate towards remyelinating plaque, we estimated the chemotactic activity of CD49d^+^CD154^+^ lymphocytes towards differentiating hOPCs through transwell membrane. We found that RR-MS, contrary to HC CD49d^+^CD154^+^ or RR-MS and HC CD49d^-^CD154^-^ lymphocytes, migrated into direction of hOPCs (Figure 2C). Flow cytometric analysis of CD49d^+^CD154^+^ lymphocytes, which passed through the transwell membrane, revealed a presence of CD3^+^ and CD19^+^ cells at a ratio 10:1 (Figure 2C).

Next, we addressed the question about the environment created during the interaction between CD49d^+^CD154^+^ lymphocytes and maturating OPCs. The analysis of the chemokines and cytokines in the co-culture supernatants of sorted RR-MS CD49d^+^CD154^+^ lymphocytes with maturing hOPCs revealed significantly higher concentrations of CCL1, CCL8, CCL20, CCL21, CCL22, CCL24, CCL25, CCL26, CCL27, CX3CL1 CXCL1, CXCL2, CXCL5, CXCL6, CXCL9, CXCL12, IL-8, TNF-α, IL-1β, IL-6, and MIF1 than in supernatants of the co-culture with HC CD49d^+^CD154^+^ lymphocytes (Figure 2B right panel and Table 2).

We also addressed a role of CXCL12/CXCR4/CXCR7 axis, which was previously proven to orchestrate the OPC recruitment and maturation during CNS regeneration [27]. We found that RR-MS after exposure to myelin peptides produced readily detectable amounts of CXCL12 (33 ± 12.0 pg/mL), whereas similarly stimulated HC PBMCs did not release this chemokine at the detectable level (<5.3 pg/mL) (Table 1, green font). In supernatants of hOPCs alone, the concentrations of CXCL12 were below the detection limit (<5.3 pg/mL), but in the co-culture with RR-MS but not HC CD49d^+^CD154^+^ lymphocytes, the concentrations of this chemokine significantly increased approximately 8–9-fold (264 ± 59.2 pg/mL) in comparison to myelin-stimulated RR-MS lymphocytes alone (Figure 2B and Table 2 green font). Flow cytometry analysis of specific receptors for this chemokine revealed that CXCR4 expression was markedly increased on the RR-MS CD49d^+^CD154^+^ lymphocyte surface after exposure to myelin peptides, while it was not affected in HC CD49d^+^CD154^+^ or CD49d^−^CD154^−^ lymphocytes (Figure 2D). Conversely, CXCR7 expression was diminished on the surface of RR-MS CD49d^+^CD154^+^ lymphocytes exposed to myelin peptides (Figure 2D). We observed no changes in the CCR6 expression, a receptor regulating diapedesis of Th17 cells into the EAE CNS [28], in any of the experimental settings (Figure 2D). Based on these experiments, we concluded that RR-MS CD49d^+^CD154^+^ lymphocytes not only increased CXCL12 production in the presence of maturing hOPCs, but also increased surface expression of CXCR4 and decreased CXCR7 competing with hOPCs for CXCL12.

CD154-CD40 dyad is considered as one of the most important immune checkpoint regulators, which is fundamental for developing of the chronic inflammation during MS. As myelin-specific lymphocytes are characterized by high CD154 expression and OPCs during maturation acquire MHC II molecules [29], we tested hOPCs for expression of CD40 (CD154 ligand), an antigen presentation costimulatory molecule. First, we showed that maturing hOPCs were characterized by increasing CD40 expression (Figure 2E, right panel). As the morphology of hOPCs changed during their polarization, the use of cytometric method allowed us to analyze different populations based on the cell volume and nuclei density (FSC vs. SSC dot plot). In the region of large cells (R1 >20 μm), we noted high CD40 expression, while in the region characterized by small cells (R2, approximately 10–20 μm), CD40 expression was low (Figure 2E, left panel). As no distinct cut point between these populations was identified, probably as a result of flowing morphological changes, we concluded that hOPCs acquire CD40 during maturation. This expression was slightly decreased in the presence of myelin-specific RR-MS CD49d^+^CD154^+^ lymphocytes but the difference was not statistically significant (Figure 2E). Next, using CSFE method, we showed that maturating hOPCs, which previously phagocytosed myelin peptides [29], stimulated RR-MS CD49d^+^CD154^+^ proliferation. This process was CD40-CD154 dependent as neutralizing of CD40 on the surfaced OPC by anti-CD40 mAbs abrogated proliferation of RR-MS CD49d^+^CD154^+^ lymphocytes (Figure 2F). Furthermore, high sCD40 concentration in the co-culture supernatants additionally confirmed CD40-CD154 interaction (Figure 2G).

### 3.3. CD49d^+^CD154^+^ Lymphocytes are Present in the EAE Mouse Brain

To confirm whether our in vitro observations are reflected in vivo in brain, we used a mouse model of MS (EAE). C57Bl/6 mice immunized with MOG_35-55_ were sacrificed and sampled for histopathology and immunohistochemistry three weeks after the disease peak (neurological signs scored for 1), which reflected remyelination period. First, using chemical staining of myelin, neuron cell bodies, and DNA, we identified pathological structure of EAE brain three weeks after the EAE peak (Figure 3A). Using hematoxylin-eosin staining, we found leucocyte presence in subventricular zone (SVZ) during EAE remission (Figure 3B). Next, using IHC with specific mAb O4/CD49d/CD154 labelling, we noted that in EAE, CD49d^+^/CD154^+^ signal was co-localized around dark areas outlined by O4^+^ signal from maturing OPCs (Figure 3C). We also found in EAE that the decrease in O4^+^ signal was accompanied by the increase in CD49d^+^ and CD154^+^ signals, while in HCs, only O4^+^ signal was detected. High co-localization of CD49d^+^ and CD154^+^ fluorescence signals in EAE mouse (average overlap coefficient >0.93) suggested that signals came from the same double-positive cells. Low co-localization of fluorescence signals of O4^+^ vs. CD49d^+^ (average overlap coefficient ~0.24) and O4^+^ vs. CD154^+^ (average overlap coefficient <0.15) indicated that the signals came from different cell populations located in close proximity (Figure 3C, right panel). MBP/CD49d/CD154 tricolor labeling revealed that CD49d^+^ and CD154^+^ signals surrounded the areas with high expression of MBP in EAE (Figure 4A,B). Similar to the experiment with O4/CD49d/CD154 labeling, high overlap coefficient was demonstrated in CD49d^+^ vs. CD154^+^ signals (0.968) (Figure 4C). Due to the methodological limitation (tricolor staining), we were not able to apply additional color labelling for antigen specificity for lymphocytes, thus we failed to clearly demonstrate that CD49d^+^/CD154^+^ signals came exclusively from lymphocytes.

## 4. Discussion

We generally hypothesized that limited remyelination after relapse of MS is associated with the presence of immune cells in remyelinating plaques, which can change the natural process of CNS regeneration by oligodendrocytes. This theory is supported by the incapability of OPCs to repopulate the area of demyelination as the effect of the presence of inhibitory factors and/or lack of stimuli required to generate remyelinating OLs in the area of demyelination [30,31].

We demonstrated that during differentiation of OPCs to mature myelin protein-producing OLs, these cells synthetize chemokines and proinflammatory cytokines, which can recruit and activate proinflammatory lymphocytes. The concentrations of proinflammatory factors in the culture supernatants were significantly augmented in the presence of RR-MS CD49d^+^CD154^+^ lymphocytes. Although both RR-MS CD49d^+^CD154^+^ lymphocytes and maturing hOPCs alone can separately produce some of them, their increased concentrations in the co-cultures cannot be assumed merely as an additive effect but rather as the result of a positive feedback regulatory loop. Moreover, we demonstrated that RR-MS CD49d^+^CD154^+^ lymphocytes in response to myelin peptides enhanced CXCR4 expression, a receptor for CXCL12, and possess CCR6, which is considered to be a marker of brain-infiltrating leukocytes [28,32]. Simultaneously, we demonstrated up-regulation of CXCL12 and CCL20 (ligand for CCR6) expression in the co-cultures of RR-MS CD49d^+^CD154^+^ and maturing OPCs. Interestingly, we noted that other chemokines belonging to CXCL family acted in a similar manner. This pattern of response was specific only for RR-MS CD49d^+^CD154^+^ lymphocytes, as CD49d^-^CD154^-^ or HC CD49d^+^CD154^+^ lymphocytes alone as well as in co-culture with maturing OPCs do not cause such changes. In adults, CXCL12/CXCR4/CXCR7 axis regulates OPC proliferation, migration, and differentiation, and its role is well-documented in RR-MS pathogenesis [27]. Physiologically, CXCL12 is not synthesized by maturating OPCs expressing only its receptors. However, the dramatic increase in CXCL12 concentration in the co-culture of maturating OPCs with RR-MS CD49d^+^CD154^+^ cells at a ratio of 50:1 and five times lower concentration of CXCL12 in the culture of RR-MS PBMCs suggest that lymphocytes cannot be the only source of this chemokine. Previous studies demonstrated CXCL12 up-regulation in MS silent and chronic plaques, especially within hypertrophic astrocytes near the plaque edge [33]. Therefore, it is rational to speculate that elevated levels of CXCL12 in the CNS might attract CXCR4-expressing lymphocytes, which compete with OPCs for CXCL12 resulting in insufficient remyelination. The polarity of CXCL12 expression in the active and inactive MS plaques changes from abluminal toward the luminal surface of the BBB vascular endothelium, which is considered to be a critical element for the trafficking of CXCR4-positive leukocytes into the perivascular and then into CNS parenchyma space [34]. In our study, we showed that RR-MS CD49d^+^CD154^+^ lymphocytes are characterized by enhanced CXCR4 expression and CXCL12 synthesis. We also demonstrated that RR-MS CD49d^+^CD154^+^ lymphocytes were characterized by low (comparable to CD49d^-^CD154^-^ cells) expression of CXCR7, a scavenger receptor that neutralizes CXCL12 within the extracellular space [35]. Physiologically, this process, mediated by reactive astrocytes, which forms CXCL12 chemokine gradient, allows neural progenitor cells to migrate towards demyelinating lesions [27]. Based on our results, we postulate that deregulation of this chemokine axis and other chemokines that belong to CXCL family by MS CD49d^+^CD154^+^ lymphocytes make plaques more vulnerable to recurrent demyelination and repeated attacks by brain-infiltrating immune cells than normal-appearing white matter. Furthermore, RR-MS CD49d^+^CD154^+^ lymphocytes impair OPC migration to the demyelinating plaque by competitive chemokine neutralization.

Another unique feature of RR-MS CD49d^+^CD154^+^ lymphocyte subpopulation, contrary to HC cells, was its capability to proliferate when RR-MS PBMCs were exposed to the myelin peptides or maturating OPCs previously preincubated with myelin peptides. Both processes were dependent on the CD154-CD40 interaction as increased sCD40 (ligand for CD154) concentrations in the culture supernatants were observed and neutralizing mAbs anti-CD40 abrogated lymphocyte proliferation. We additionally revealed increased expression of CD40 costimulatory molecules within OPC maturation. It is in line with previous data, which presented the expression of MHC II on maturing OPCs and activation of memory/effector CD4^+^ cells via MHC II by OPCs, which phagocytosed myelin peptides [29]. These evidences point to the possibility that maturating OPCs can prolong inflammation acting as antigen-presenting cells interacting with lymphocytes and mediating their proliferation. As CD19^+^ lymphocytes were strongly attracted by maturating OPCs and were present in the EAE brain during remyelination, it is possible that CD19^+^ cells are also engaged in the induction of MS myelin-specific lymphocytes [36].

To translate our in vitro findings into in vivo status during remyelination, an EAE mouse model was employed. We demonstrated that CD49d^+^ and CD154^+^ signals are co-localized within the areas with expression OPC marker, O4, and high expression of MBP, which collectively suggest presence of CD49d^+^CD154^+^ cells in the remyelinating plaques in EAE. The limitation of our study is inability to clearly demonstrate that CD49d^+^CD154^+^ cells present in the EAE mouse brains were lymphocytes. Additional experiments using intravital two-photon imaging [37] may help to confirm crossing BBB by CD49d^+^CD154^+^ lymphocytes and their proliferation within remyelinating plaques.

The newly emerging role of myelin-specific CD49d^+^CD154^+^ lymphocytes in RR-MS pathogenesis is further outlined in our previous study demonstrating that these cells affected OPC maturation and function. OLs generated in the presence of myelin-specific CD49d^+^CD154^+^ lymphocytes were characterized by imbalanced MBP and PLP production [38]. Therefore, in RR-MS, OPC maturation and function in response to CNS injury is disturbed by autoreactive lymphocytes resulting in inefficient remyelination during disease remission. In this study, we demonstrated possible mechanisms promoting induction of myelin-specific CD49d^+^CD154^+^ lymphocytes both in the periphery as well as in the CNS.

## Figures and Tables

**Figure 1 cells-09-00015-f001:**
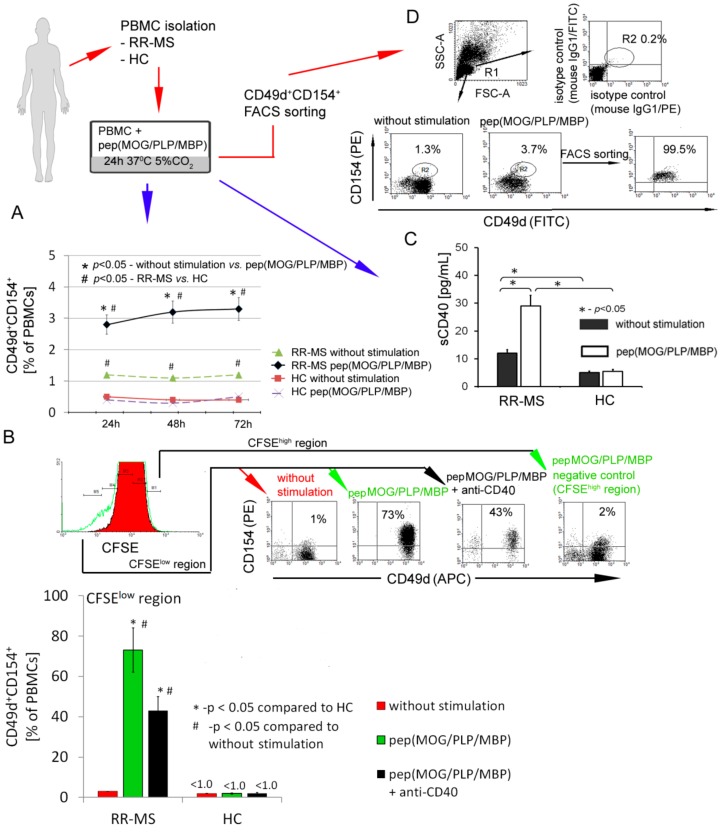
CD49d^+^CD154^+^ lymphocytes of relapse remitting multiple sclerosis (RR-MS) patients proliferate in vitro in response to myelin peptides. (**A**) RR-MS contrary to healthy control (HC) peripheral blood mononuclear cells (PBMCs) characterized by higher percentage of CD49d^+^CD154^+^ lymphocytes. Additionally, this observed difference is amplified in incubation of RR-MS PBMCs with myelin proteins. Data are presented as means ± SD from independent experiments in RR-MS n = 10 and HC n = 10. (**B**) Proliferation of RR-MS CD49d^+^CD154^+^ lymphocytes induced by myelin peptides was dependent on CD40-CD154 interaction. Data are presented as means ± SD from independent experiments in RR-MS n = 4 and HC n = 4, and the example of flow cytometry analysis from one of four independent experiments. Proliferating cells gated in CFSE^low^ region were characterized by expression of CD49d and CD154 receptors. Cells gated in CFSE^high^ region (non-proliferating cells) were negative for CD49d and CD154 receptors. (**C**) RR-MS PBMCs stimulated by myelin peptides released sCD40 into supernatants. Data are presented as means ± SD from independent experiments in RR-MS n = 10 and HC n = 10. (**D**) The representative example of flow cytometry analysis of RR-MS lymphocyte purity isolated by FACSAria sorting. These subpopulations were used for further experiments.

**Figure 2 cells-09-00015-f002:**
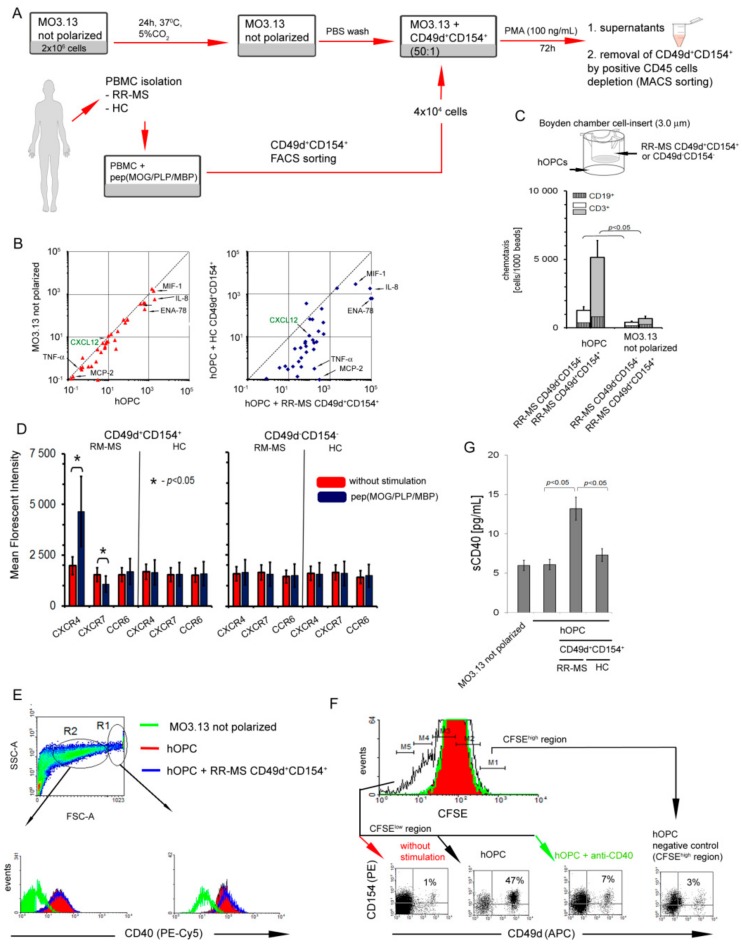
Interaction of RR-MS CD49d^+^CD154^+^ lymphocytes with maturing human oligodendrocyte precursor cells (hOPCs) generates positive proinflammatory feedback loop. (**A**) Human MO3.13 cells stimulated by phorbol 12-myristate 13-acetate (PMA), as the cellular model of OPC polarization to mature oligodendrocytes (OLs), and RR-MS CD49d^+^CD154^+^ lymphocytes induced from myelin peptide-stimulated RR-MS PBMCs were used for chemotaxis and co-culture experiments. (**B**) hOPCs released chemoactive factors (left panel), of which concentrations are amplified in the presence of RR-MS contrary to HC CD49d^+^CD154^+^ lymphocytes (right panel). Data are presented as means ± SD from independent experiments in RR-MS n = 10 and HC n = 10. (**C**) RR-MS CD49d^+^CD154^+^, opposite to CD49d^-^CD154^-^ lymphocytes, were more intensively recruited by hOPCs. RR-MS lymphocytes were CD3^+^ and CD19^+^ cells (T:B 10:1). Bars are presented as means ± SD calculated from independent experiments in RR-MS n = 4. (**D**) During myelin peptide-induced proliferation of RR-MS CD49d^+^CD154^+^ lymphocytes, they acquired CXCR4 and lost CXCR7. Bars are presented as means ± SD from independent experiments in RR-MS n = 4 and HC n = 4. (**E**) hOPCs during maturation overexpressed CD40. As the morphology of MO3.13 cells during polarization was not homogenous, two regions (R1 and R2) were analyzed. The example of flow cytometry analysis of one of four independent experiments. (**F**) pepMOG/proteolipid protein (PLP)/myelin basic protein (MBP)-treated hOPCs cultured with RR-MS PBMCs stimulated proliferation of CD49d^+^CD154^+^ subpopulation, dependent on CD40-CD154 interaction. The example of flow cytometry analysis of one of four independent experiments. (G) Co-culture of RR-MS CD49d^+^CD154^+^ lymphocytes with hOPCs resulted in increased sCD40 concentration in supernatants. Data are presented as means ± SD from independent experiments in RR-MS n = 10 and HC n = 10.

**Figure 3 cells-09-00015-f003:**
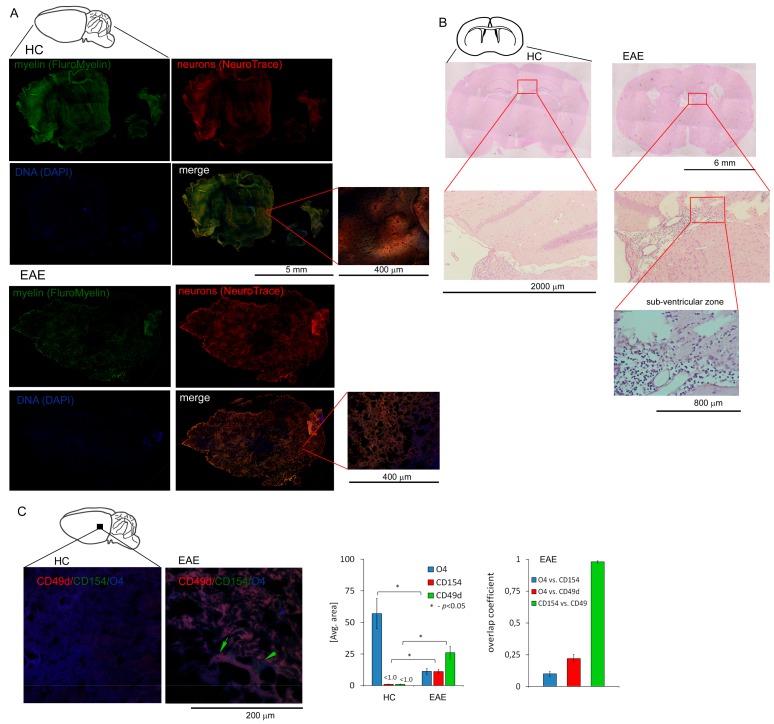
Presence of CD49d^+^CD154^+^ cells in the experimental autoimmune encephalomyelitis (EAE) brain three weeks after the disease peak. (**A**) The example of histopathological examination of sagittal brain slices stained for myelin, neuron cell bodies, and DNA demonstrating the regions deficient in myelin and neurons in the EAE brain (n = 4) three weeks after the disease peak vs. HC brain (n = 4) (**B**) The example of histopathological examination (EAE n = 4, HC n = 4) of horizontal brain section demonstrated the presence of inflammatory cells three weeks after EAE peak in the sub-ventricular zone (SVZ). ((**C**), left panel) IHC microscopy analysis of sagittal brain slices revealed high CD49d^+^ (red pseudocolor) and CD154^+^ (green) signals which were co-localized around maturating OPCs (labelled by O4, blue pseudocolor) during remission. Green arrows point the regions with high double-positive signal for CD49d^+^ and CD154^+^ close to the maturating OPCs (O4^+^). ((**C**), right panel) O4, CD49d, and CD154 fluorescence intensity and overlap coefficient analysis. Nonspecific fluorescence (signal noise) was electronically diminished to the level when nonspecific signal was undetectable (background). The bars represent average fluorescence expression ± SD from four independent experiments (one representative mouse per experiment).

**Figure 4 cells-09-00015-f004:**
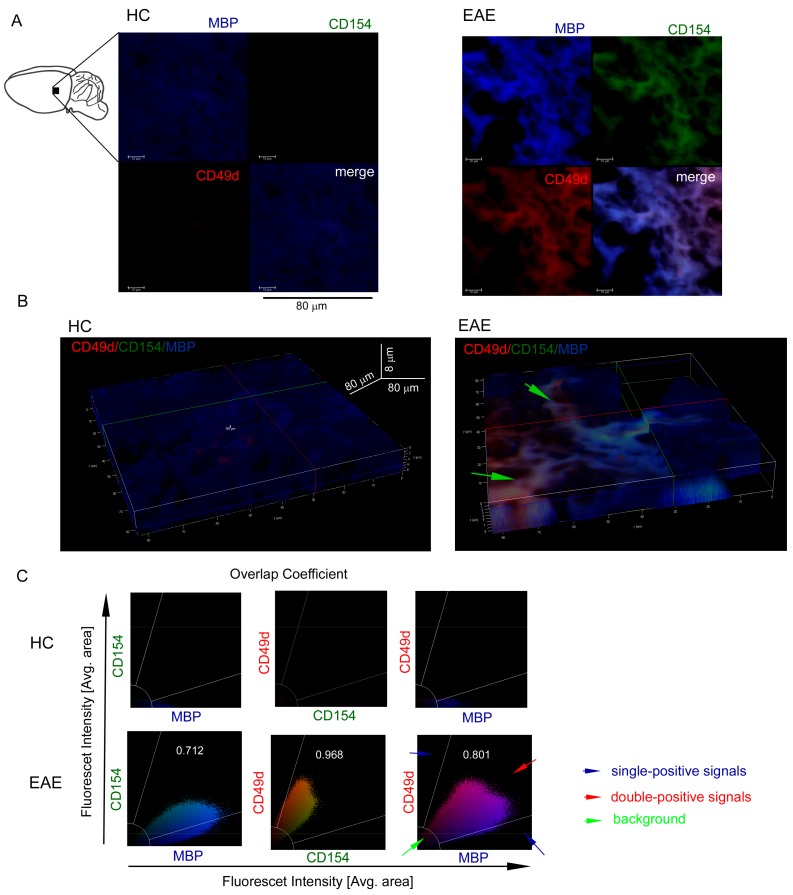
CD49d^+^CD154^+^ cells in the EAE brain three weeks after the disease peak are localized close to the remyelination lesions. (**A**) Immunohistochemistry (IHC) microscopy analysis of sagittal brain slices revealed high MBP^+^ (blue pseudocolor) signal three weeks after the EAE peak (upper-left picture) confirming remyelination. High MBP^+^ expression was accompanied by the high CD49d^+^ (red pseudocolor) and CD154^+^ expressions (green pseudocolor). The example of one of four independent experiments (EAE n = 4, HC n = 4). (**B**) Z-stack analysis in 3D projection demonstrated that CD49d^+^/CD154^+^ signals were co-localized around the regions with high MBP^+^ signal. Green arrows point the regions with high double-positive signal for CD49d^+^ and CD154^+^ close to the regions with high MBP expression. The example of one of four independent experiments (EAE n = 4, HC n = 4). (**C**) The example of overlap coefficient analysis (EAE n = 4, HC n = 4). Cytofluorograms of CD154 vs. MBP, CD49d vs. CD154, and CD49d vs. MBP signals from the EAE brain confirmed high expression of MBP with co-expression of CD49d^+^CD154^+^ cells.

**Table 1 cells-09-00015-t001:** RR-MS PBMCs stimulated by myelin oligodendrocyte glycoprotein (MOG)/PLP/MBP peptides opposite to HC produced proinflammatory cytokines and chemokines. Cytokines were measured in PBMC supernatants during CD49d^+^CD154^+^ lymphocyte expansion (PBMC+pep/MOG/PLP/MBP) with the use of Bio-Plex Pro™ Human Chemokine Assays. Data are presented as means ± SD from ten independent experiments.

	PBMC (2 × 10^6^ cells/mL)	
	RR-MS (n = 10)	HC (n = 10)
CCL1 (I-309)	22 ± 14.2 ^#^	6.8 ± 2.99
CCL2 (MCP-1)	552 ± 177.2 ^#^	66 ± 25.9
CCL3 (MIP-1α)	<3.1	<3.1
CCL7 (MCP-3)	134 ± 69.2 ^#^	<3.2
CCL8 (MCP-2)	57 ± 23.5 ^#^	<0.6
CCL11(Eotaxin)	34.8 ± 18.66	35.4 ± 13.33
CCL13 (MCP-4)	<9.4	<9.4
CCL15 (MIP-1 D )	<3.1	<3.1
CCL17 (TARC)	27 ± 10.9 ^#^	<3.4
CCL19 (MIP-3β)	<9.1	<9.1
CCL20 (MIP-3α)	26 ± 9.3 ^#^	<8.7
CCL21 (6Ckine)	69 ± 24.9 ^#^	<1.2
CCL22 (MDC)	124 ± 58.3 ^#^	<5.9
CCL23 (MPIF-1)	<3.5	<3.5
CCL24(Eotaxin-2)	369. ± 81.1 ^#^	218 ± 67.1
CCL25 (TECK)	127 ± 39.0 ^#^	50.8 ± 25.1
CCL26 (Eotaxin-3)	9.2 ± 2.19 ^#^	<1.4
CCL27 (CTACK)	39.4 ± 8.31 ^#^	0.8 ± 0.42
CX3CL1 (Factalkine)	109 ± 31.7 ^#^	<9.6
CXCL12 (SDFα+β)	33 ± 12.0 ^#^	<5.3
CXCL1 (Gro-α)	456 ± 119.34 ^#^	46.1 ± 27.11
CXCL2 (Gro-β)	153 ± 33.9 ^#^	12 ± 7.8
CXCL5 (ENA-78)	6205 ± 1842.3 ^#^	676 ± 227.4
CXCL6 (GCP-2)	18.9 ± 10.34 ^#^	<3.1
CXCL8 (IL-8)	11206 ± 909.4 ^#^	867 ± 299.7
CXCL9 (MIG)	87 ± 44.7 ^#^	<4.9
CXCL10 (IP-10)	<2.6	<2.6
CXCL11 (I-TAC)	59 ± 17.9 ^#^	<2.9
CXCL13 (BCA-1)	<0.1	<0.1
CXCL16 (SCYB16)	76 ± 17.8 ^#^	<13.4
TNF-α	39 ± 16.6 ^#^	<3.1
IFN-γ	28.1 ± 3.2 ^#^	<0.1
IL-1β	14 ± 3.9 ^#^	5 ± 3.7
IL-2	3.0 ± 1.96 ^#^	<0.4
IL-4	7.1 ± 5.77	5.9 ± 3.08
IL-6	28 ± 8.1 ^#^	8 ± 3.2
MIF1	1071 ± 377.0 ^#^	408 ± 144.2
IL-10	<5.1	<5.1
IL-16	219 ± 83.1	213 ± 57.2
GM-CSF	<1.5	<1.5

#—statistically significant differences between RR-MS and HC PBMC. Red font—CXC type chemokines (associated with recruitment of neutrophils and lymphocytes). Blue font—CC type chemokines (associated with recruitment of lymphocytes, monocytes, mast cells, and eosinophils). Green font—chemokines which regulate neuronal-glial interactions and development of CNS.

**Table 2 cells-09-00015-t002:** Maturing OPCs produced chemokines and proinflammatory cytokines. RR-MS opposite to HC CD49d^+^CD154^+^ lymphocytes together with maturating OPC created proinflammatory circumstances, which accelerated inflammation in feedback reaction. Cytokines were measured in supernatants of maturating OLs and their co-culture with sorted CD49d^+^CD154^+^ lymphocytes by the use of Bio-Plex Pro™ Human Chemokine Assays. Data are presented as means ± SD from independent experiments in RR-MS n = 10 and HC n = 10.

	MO3.13 (2 × 10^6^ cells/mL)			
	Not Polarized	Polarized to OLs	Polarized to OLs + RR-MS CD49d^+^CD154^+^	Polarized to OLs + HC CD49d^+^CD154^+^
CCL1 (I-309))	3.6 ± 2.21	6.8 ± 2.99	68.1 ± 18.06 * ^§^	10.4 ± 4.94
CCL2 (MCP-1)	391.6 ± 94.32	895 ± 209.9†	977 ± 494.1	927 ± 200.3
CCL3 (MIP-1α)	<3.1	<3.1	<3.1	<3.1
CCL7 (MCP-3)	<3.2	<3.2	<3.2	<3.2
CCL8 (MCP-2)	<0.6	<0.6	62 ± 15.9 * ^§^	<0.6
CCL11(Eotaxin)	64.7 ± 19.57	95.6 ± 26.47	137.0 ± 31.59 * ^§^	81.8 ± 21.60
CCL13 (MCP-4)	<9.4	<9.4	<9.4	<9.4
CCL15 (MIP-1 D )	<3.1	<3.1	<3.1	<3.1
CCL17 (TARC)	<3.4	<3.4	<3.4	<3.4
CCL19 (MIP-3β)	<9.1	<9.1	<9.1	<9.1
CCL20 (MIP-3α)	<8.7	<8.7	99 ± 18.7 * ^§^	<8.7
CCL21 (6Ckine)	18 ± 4.9	29 ± 10.7	76.5 ± 18.2 * ^§^	26 ± 12.94
CCL22 (MDC)	<5.9	<5.9	23 ± 12.1 * ^§^	<5.9
CCL23 (MPIF-1)	<3.5	<3.5	<3.5	<3.5
CCL24(Eotaxin-2)	<1.1	<1.1	71 ± 27.1 * ^§^	12 ± 11.9*
CCL25 (TECK)	46 ± 18.5	69 ± 23.3	263 ± 83.3 * ^§^	79. ± 20.8
CCL26 (Eotaxin-3)	<1.4	<1.4	16.8 ± 2.60 * ^§^	<1.4
CCL27 (CTACK)	0.4 ± 0.33	0.5 ± 0.37	29.5 ± 15.19 * ^§^	1.5 ± 1.25
CX3CL1 (Factalkine)	57 ± 10.48	89 ± 11.8†	506 ± 46.4 * ^§^	79 ± 30.9
CXCL12 (SDFα+β)	<5.3	<5.3	264 ± 59.2 * ^§^	<5.3
CXCL1 (Gro-α)	208.1 ± 79.34	347.5 ± 127.78†	1533.4 ± 301.68 * ^§^	365.9 ± 195.19
CXCL2 (Gro-β)	7 ± 3.9	21 ± 8.8†	217.9 ± 185.66 * ^§^	33 ± 13.5
CXCL5 (ENA-78)	296. ± 164.8	862 ± 227.4†	10,432 ± 1129.3 * ^§^	985 ± 395.2
CXCL6 (GCP-2)	64.8 ± 23.95	54.9 ± 27.33	329.9 ± 105.99 * ^§^	64.0 ± 28.58
CXCL8 (IL-8)	913. ± 209.0	3235 ± 998.6†	16,896 ± 5981.1 * ^§^	3626 ± 672.1
CXCL9 (MIG)	<4.9	<4.9	69 ± 13.9 * ^§^	<4.9
CXCL10 (IP-10)	<2.6	<2.6	<2.6	<2.6
CXCL11 (I-TAC)	<2.9	<2.9	<2.9	<2.9
CXCL13 (BCA-1)	<0.1	<0.1	<0.1	<0.1
CXCL16 (SCYB16)	360 ± 178.2	383 ± 166.1	388 ± 258.7	342 ± 187.0
TNF-α	<3.1	<3.1	36.5 ± 12.22 * ^§^	<3.1
IFN-γ	<0.1	<0.1	<0.1	<0.1
IL-1β	<0.3	<0.3	25.8 ± 12.33* ^§^	<0.3
IL-2	<0.4	<0.4	<0.4	<0.4
IL-4	<0.4	<0.4	<0.4	<0.4
IL-6	3 ± 1.8	24 ± 9.8†	259 ± 13.2 * ^§^	27 ± 10.2
MIF1	1931 ± 822.2	2735 ± 767.9†	3687 ± 872.8 * ^§^	2708. ± 772.4
IL-10	<5.1	<5.1	<5.1	<5.1
IL-16	<13.7	<13.7	<13.7	<13.7
GM-CSF	<1.5	<1.5	<1.5	<1.5

†—statistically significant differences between MO3.13 and MO3.13 polarized to OLs; *—statistically significant differences between MO3.13 polarized to OLs and MO3.13 polarized to OLs with CD49d^+^CD154^+^ cells; §—statistically significant differences between MO3.13 polarized to OLs with RR-MS CD49d^+^CD154^+^ and MO3.13 polarized to OLs with HC CD49d^+^CD154^+^ cells. Red font—CXC type chemokines (associated with recruitment of neutrophils and lymphocytes). Blue font—CC type chemokines (associated with recruitment of lymphocytes, monocytes, mast cells and eosinophils). Green font—chemokines which regulate neuronal-glial interactions and development of CNS.

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
