# Peer review of "Multiple Sclerosis CD49d+CD154+ As Myelin-Specific Lymphocytes Induced During Remyelination"

_cells, 2019, doi:10.3390/cells9010015_

Round 1
Reviewer 1 Report
I red the article by Piatek and colleagues. While I think this is an interesting finding, it is not sufficiently convincing to me, especially the animal experiments. My second main concern is the clarity of the manuscript, mainly abstract and introduction (including English grammar). I can thus not recommend publication in its present form. Please find more detailed comments below.
Abstract:
Should be simplified, not to easy to read, e.g. “The co-culture of maturing OPCs with myelin-specific CD49d+CD154+ lymphocytes, revealed the increased secretion of proinflammatory chemokines/cytokines enhancing the presence of lymphocytes within remyelinating plaque” – this is done in culture, but how does this relate to the remyelinating plaque?
In the second part of the abstract, you elaborate on OLs: “We confirmed in vivo the presence of CD49d+CD154+ lymphocytes close to OLs…” – this is not a confirmation of the OPC findings or why do you now mention OLs?
Introduction:
“The main reason of slowly progressing, irreversible disease-related disability is a renewal of only fragmentary tissue” – I do not understand this sentence. You mean regeneration of injured tissue? And how would that be the reason for progression?
EAE can also be progressive (depending on species and induction protocol), not only RR.
“The significance of CD154-CD40 dyad in the disease progression has been demonstrated in CD154, CD40 knock-out mice, CD40 single-nucleotide polymorphism in MS patients, as well as with the use of neutralizing mAbs anti-CD154 (clone IDEC-131) and anti-CD40 in human and EAE, respectively” – this sentence is very long and not readily understandable. The significance of these cells has been shown in CD40 SNP? This does not make sense grammar wise. Last part: “… in human and EAE” . you mean: in MS and EAE?
“Oligodendrocytes (OLs), as the principle cells responsible for neuron myelin sheath formation” – the term neuron myelin sheath makes no sense.
Please clearly define your study niche in the introduction: why is your study important, what does it address that has not been addressed before?
Remove result summary from the last introduction part.
Overall, the introduction is too long and contains to much information not needed to read this article. Please condensate the introduction and improve clarity.
Methods:
Age: mean? Median? What is the IQR?
Add measure of variance to other parameters, eg. disease duration.
How did you match controls: propensity weighted?
“The BBB was damaged…” – odd formulation, just state how you applied pertussis toxin.
State how many animals you used in total for your study.
Why did you perfuse with GA and not with FA? I am a bit puzzled by the fact that you were able to run IHC for these complex antigens in GA-fixed tissue – normally GA abolishes any antibody-binding site and is mainly used for electron microscopy.
“fixed in 4% PFA…” - This is written in many papers but it is strictly speaking not correct – you do not use PFA for fixation; this does not fix tissue since it is the inactive component. Formaldehyde is the active fixing component.
Did the Cd49/154 lymphocytes from controls also react to myelin stimulation?
Were you blinded during analysis? Were the experiments randomized whenever possible?
Please make your report in accordance with the ARRIVE guidelines for animal experimentation (Kilkenny et al., 2010).
Figures:
Check your axis labels and figure legends for mistakes, eg. 2D – “Mean Florescent Intencity”
Figure 3: in both the results section and the figure legend, you refer to remyelinating plaques – how do you know that this plaque is remyelinated/-ing? You did not include any myelin antigen to your IHC and no histology staining like Eriochrome or LFB in order to say anything about the myelin status of the lesions. Moreover, figure 1A is completely out of context – is that a brain lesion? If yes, where in the brain? Also, the IHC images in figure 3 are not convincing. I assume these are sagittal brain slices for the O4 staining? There seems to be a lot of background in these images; since you expect O4 labeling OLs, you would expect a more intense staining in the white matter which does not seem to be the case here. Again, the higher mag images of 49/154 look like background to me (just a homogenous brownish staining in the whole tissue). There is also no kind of quantification supporting the claim of spatial proximity of these lymphocytes to OLs. And again: why is it now OLs since you elaborated on OPCs the preceding paragraphs. Why did you not include an OPC marker like PDGRFalpha to your IHC?
Author Response
Reviewer 1
According to Reviewer’s suggestions we have improved Abstract and Introduction sections, in particular, the sentences mentioned in the Review. Methodology paragraph was also corrected according to all aspects risen by the Reviewer.Below, you find the answers to the most important aspects:
Although perfusion with FA is widely used, treatment with GA is also practiced and recommended (abcam.com>protocols>ihc-fixation-protocol). In our previous studies, we have successfully used GA to visualize miRNA processing proteins without its significant interference on specific antibody-binding in the brain [Lewkowicz et al.].Lewkowicz, P.; Cwiklińska, H.; Mycko, M.P; Cichalewska, M.; Domowicz, M.; Lewkowicz, N.; Jurewicz, A.; Selmaj, K.W. Dysregulated RNA-Induced Silencing Complex (RISC) Assembly within CNS Corresponds with Abnormal miRNA Expression during Autoimmune Demyelination. J Neurosci. 2015; 35(19):7521-7537.
We agree that the statement ‘fixed in 4% PFA’ is not correct. We have changed it for ‘fixed in 4% PFA solution’as paraformaldehyde in solution depolymerizes into formaldehyde. In the Figure 1A we have shown RR-MS HC CD49d+CD154+ lymphocytes with and without stimulation by myelin peptides. Data are presented as the percentage of PBMCs. We did not use randomization in this study. MS patients were consecutively recruited to the study. ICC and IHC quantitative analysis was performed by two independent investigators, prof. M. Wieczorek and dr. S. Michlewska, who were blinded for the sample allocation.‘ According to the suggestion, we have added the report prepared in accordance with ARRIVE guidelines for animal experiments.
Figures:
According to the Reviewer’s suggestion, we have improved Figure 1 and Figure 2, and included more data to the Figure 3 to confirm the presence of CD49d+CD154+ lymphocytes in the brain during disease remission. The statement ‘presence of CD49d+CD154+ lymphocytes within remyelinating plaques’ was replace with: ‘Presence of CD49d+CD154+ lymphocytes in EAE brain three weeks after the disease peak’ In the Figure 3A, we have added new photographs from microscopy analysis with different magnifications demonstrating inflammatory cell infiltration in EAE brain during disease remission. We have also performed statistical analysis of immunofluorescence intensity, and calculated the overlap coefficient factor which additionally confirmed that signal from CD49d and CD154 belongs to the same cells (Figure 3C). The non-specific signal was eliminated by subtracting signal from the background. Additional information has been added to the Material and Method: (‘Fluorescence intensity was determined as the average fluorescence (Avg. area), the sum of the fluorescence from all segments divided by the number of segments (IPLab Pathway 4.0 software, BD Bioscience). The average fluorescence for each pseudocolor was calculated using 31 single regions (Region Of Interest - ROI) taken from four independent experiments (one slide per experiments). The level of baseline fluorescence was established individually for each experiment. Nonspecific fluorescence (signal noise) was electronically diminished to the level when nonspecific signal was undetectable [23]’). A new paragraph describing these results has been added to the Result section ‘.
3.3. CD49d+CD154+ lymphocytes are present in the EAE mouse brain
‘To confirm whether our in vitro observations are reflected in vivo in brain, we used mouse model of MS (EAE). Firstly, using IHC, we confirmed that leucocytes were present in the EAE brain during remyelination period three weeks after the disease peak (Figure 3A). Secondly, we noted that in EAE, opposite to HC brains, CD49d/CD154 signal was co-localized around dark areas outlined by O4 signal of maturing OPCs (Figure 3B). We also found in EAE that the decrease in O4 signal was accompanied by the increase in CD49d+ and CD154+ signals, while in HCs only O4 signal was detected (Figure 3C left panel). High co-localization of CD49 and CD154 florescence signals in EAE mouse (average overlap coefficient factor >0.93) suggested that signals came from the same double-positive cells. Low co-localization of florescence signals of O4 vs. CD49d+ (average overlap coefficient factor ~0.24) and O4 vs. CD154+ (average overlap coefficient factor <0.15) indicated that the signals came from different cell populations located in close proximity (Figure 3C right panel).’
We used O4 labeling as an antigen present on the surface of oligodendrocyte progenitors. ‘O4 has been commonly used as the earliest recognized marker specific for the oligodendroglial lineage. The monoclonal antibodies A2B5, O4, and O1 are frequently used to define distinct stages in the maturation of oligodendrocyte progenitors. In general, A2B5+/O4- defines the most immature oligodendrocyte precursors, O4+/O1- defines intermediate precursors, and O1+ defines oligodendrocytes at a more mature stage.’ (https://www.rndsystems.com/target/oligodendrocyte-marker-o4 [1-8]. We agree that the use of nestin or PDGFR labelling would even better show remyelinating areas. This aspect was risen in our second manuscript sent to Cells (Number 632631 ‘MS CD49d+CD154+ Lymphocytes Reprogram Oligodendrocytes Into Immune Reactive Cells Affecting CNS Regeneration’). Schachner, M. et al. (1981) Dev. Biol. 83:328. Bansal, R. et al. (1989) J. Neurosci. Res. 24:548. Bansal, R. and Pfeiffer, S.E. (1989) Proc. Natl. Acad. Sci. USA 86:6181. Gard, A. et al. (1995) Dev. Biol. 167:596. Reynolds, R. and Hardy, R. (1997) J. Neurosci. Res. 47:455. Ono, K. et al. (1997) J. Neurosci. Res. 48:212. Pang, Y. et al. (2000) J. Neurosci. Res. 62:510. Cai, Z. et al. (2001) Brain Res. 898:126.Reviewer 2 Report
This manuscript is aimed at resolving important issues of myelin and neuronal function repair and axonal regeneration in MS. The authors utilized array of approaches and techniques includingPBMC sampling from humans, in vitro co-culture experiments, cytokine/chemokine assays, EAE experiments in mice, immunohistochemistry. Taken into account the complexity of approaches, manuscript is well written and data presented and discussed appropriately. Introduction section and overall manuscript would be strengthened by commenting on significance of axonal and oligodendroglial damage and demise in MS and EAE, right at the beginning of the manuscript. Data from this study are likely to further understanding of immune cellular and molecular mechanisms relevant for axonal, neuronal and oligodendroglial repair in MS.

Author Response
According to Reviewer’s, suggestions we have improved and strengthened our article commenting on significance of axonal and oligodendroglial damage, particularly in the introduction section, for better understanding of immune cellular and molecular mechanisms relevant for axonal, neuronal and oligodendroglial repair in MS.
‘Multiple sclerosis (MS) is an autoimmune lymphocyte-dependent demyelinating disease of the central nervous system (CNS). Axonal loss and functional disability in forming myelin sheaths by oligodendrocytes (OLs) are two important processes responsible for the disease progression and patient death within years [1]. Relapsing-remitting MS (RR-MS) is the most prevalent form affecting mainly young people, which is characterized by acute attacks (relapses) followed by a period of partial withdrawal of symptoms (remission) [2]. Relapse mainly involves the brain areas previously affected by the disease where gradual, extensive processes of demyelination and irreversible neuron impairment had already taken place [3]. Slow but constant progression of the disease is caused by inefficient and only partial regeneration of the myelin sheath around the neuronal axons after primary attack or relapse [4], what still remains the unsolved aspect of MS pathology. This might be a result of incomplete clearance of inflammatory cells from the brain during disease remission or even proliferation of autoreactive cells within remyelinating plaques. Alternatively, autoreactive myelin-specific lymphocytes might migrate from the periphery to the CNS, preferably to the areas of previous demyelination [5]’.
Reviewer 3 Report
Background:
The authors of this paper have identified a population of CD49+ve/CD154+ve lymphocytes, these of which they claim play a role in the remyelination of the damaged CNS via their triggering the expansion and differentiation of oligodendrocyte progenitor cells. They show this with both ex-vivo experiments using PBMCs derived from Relapse-Remitting Multiple Sclerosis (MS) and healthy blood samples, as well as with in-vivo data derived from a well-established mouse MS model (MOG-peptide induced EAE). As mentioned in the abstract, a lot of research has focused upon the role of autoreactive lymphocytes in CNS demyelination but much less is known about the remyelination process.
Summation:
The subject is interesting. The ex-vivo studies are done very nicely (with some comments). The EAE data is sparse, and from the detailed materials and methods explanation in comparison to the single figure provided in figure three, I assume that the mouse work has been cut back substantially without explanation. The materials and methods seems to me much more detailed than that what relates to the results section suggesting that this is a poorly reworded fix of a larger paper. With revision I believe this paper is worthy of publication in Cells.
Points:
Introduction
Line 43: This indeed might be my personal lack of understanding this term in this context, but what is meant by “fragmentary tissue”? If not a commonly known term, please can this be re-worded?
Line 52+: “The significance of CD154-CD40 dyad...has been demonstrated…” What were the overall findings? You failed to describe them in any detail.
Materials and Methods
Line 82: [mean?] aged 43.2 ?
Line 84: (and in some other parts of the text) you use the notation 5,4 years (comma), which I presume is 5.4 years (dot). For consistency of the text please use the standard “dot” notation here and elsewhere.
Line 126: This mistake is repeated numerous times throughout the article. ‘….a mixture of protelipid protein…” It is not proteolipid protein, but a peptide sequence derived from PLP. Please correct this here and elsewhere. Call these protein-derived peptides or some other alternative name that makes it clear that the stimulating antigens are not full proteins.
Line 145: ddPCR. I saw no clear reference to mRNA quantification in the paper. Either it was performed (e.g. Fig. 2B) but without describing the results or it is not found in the paper. Either way, please fix.
RESULTS
Section 3.1 Incubated MS and Control PBMCs with a cocktail of myelin peptides (not PROTEINS) and showed an expansion of CD49+Cd154+ lymphocytes. There were more of these cells in the MS PBMCs compared to healthy controls and these displayed accelerated growth under stimulation with myelin peptides. CFSE staining/FACS of RR-MS samples demonstrated accelerated cell division after stimulation
Major Comment:
In the materials and methods, you describe the use of 10 RR-MS and 10 healthy control patients. However here in the results you provide results of a single set of MS and control PBMCs. Was the study done on individual patient or pooled PBMCs? Individual patient data would be preferred approach, as you will get information regarding stochastic differences between patients/controls. You provide error bars in Fig. 1A & C. Is his is the data from individual patients +/- St. Dev? What about Figure 1B? This is CSFE data from a single patient? Please can this be clarified in the text where all the patient data is used or representative data is used, and if the error bars represent repeat measurements of pooled PBMCs or of individual PBMCs. This is not at all clear. If indeed we are looking at 10 data points per MS and per control sample, in addition to figure 2B, it might be nice to see how the % of CD
Vertical axis for Figure 1A: Shouldn’t it read % CD49D+CD154+ cells of PBMCs?
Line 220 (etc): myelin proteins should read myelin protein-derived peptides.
Line 233: Here the investigators measured a number of chemokines and cytokines from RR-MS and control PBMCs stimulated with myelin antigens. Here it is not mentioned how this was done. Was it by multiplex immunoassay? Please make it clear. Also please make it clear if this was for pre-purified CD49+/CD154+ cells or PBMCs that were not purified? Finally we return to the question if this is pooled PBMCs from many patients or that of average results from 10 individuals in each group. None of this is made clear in the text.
Section 3.2
Here the author co-cultured a human olidodendrocte precursor cell line MO3.13 before/after maturation together with CD49+/CD154+ lymphocytes and looked for changes in cytokine/chemokine profiles as a result of the co-cultures. Also seen a
Figure 2B and Table 2. Are these the same datasets? I can’t see how the points in the plotes in Figure 2B relate to Table 2. For example CXCL12 – a very important factor that is discussed in this paper has different values in Table 2 vs. Figure 2B. Also the vertical axis for the right panel in Figure 2B represents hOPCs + healthy control CD49D/CD154+ cells is not labelled. The data looks really interesting. It just isn’t presented clearly.
Section without a heading – EAE studies.
A single figure is presented here relating to EAE mice at 2 weeks after EAE induction (disease peak). It provides a single set of slides from a single EAE mouse vs. a single control. The disease stage for this mouse is not given. Also no data relating to the histology of the remyelenating EAE brain is shown at the time of disease remission. This section is clearly the poor point of this article. Either the EAE data should be removed or
Abstract:
Line 25: …… enhancing the presence of lymphocytes within remyelinating plaque.
This data just was not shown (I assume from EAE studies).
Author Response
We have recently sent to Cells two manuscripts demonstrating a role for newly discovered CD49+CD154+ lymphocytes in MS and EAE (#632583, ‘Multiple Sclerosis CD49d+CD154+ As Myelin-Specific Lymphocytes Induced During Remyelination’ and #632631 ‘MS CD49d+CD154+ Lymphocytes Reprogram Oligodendrocytes Into Immune Reactive Cells Affecting CNS Regeneration’). Therefore, a part of Methods from the animal studies was mistakenly taken from the second manuscript. According to Reviewer’s suggestions we have improved Abstract and Introduction sections, in particular, the sentences mentioned below.
Abstract:
‘We confirmed in vivo the presence of CD49d+CD154+ lymphocytes close to maturating OPCs during disease remission in the MS mouse model (C57Bl/6 mice immunized with MOG35-55) by immunohistochemistry. Three weeks after an acute phase of experimental autoimmune encephalomyelitis, CD49d/CD154 signal was co-localized within oligodendrocyte progenitors (marked by O4)’.
Introduction:
‘Relapse mainly involves the brain areas previously affected by the disease where gradual, extensive processes of demyelination and irreversible neuron impairment had already taken place [3]. Slow but constant progression of the disease is caused by inefficient and only partial regeneration of the myelin sheath around the neuronal axons after primary attack or relapse [4], what still remains the unsolved aspect of MS pathology. This might be a result of incomplete clearance of inflammatory cells from the brain during disease remission or proliferation of autoreactive cells within remyelinating plaques. Alternatively, autoreactive myelin-specific lymphocytes might migrate from the periphery to the CNS, preferably to the areas of previous demyelination [5]’. ‘The significance of CD154-CD40 dyad in the disease progression has been demonstrated in the studies with CD154 and CD40 knock-out mice, as well as with the use of neutralizing mAbs anti-CD154 or anti-CD40 in EAE [7, 10-14]. The role of CD154-CD40 dyad was also confirmed in human studies. The single nucleotide polymorphism (SNP) analysis revealed that the variant of CD40 gene (rs 1883822C- > T) was associated with an increased risk for MS in comparison to healthy individuals about 1.5-fold in heterozygous and 2.5-fold in homozygotes, respectively [15, 16]. It was also demonstrated that stimulation of CD40 receptor on B cells of RR-MS patients resulted in significantly higher proliferation than in healthy subjects [17], and application of neutralizing anti-CD154 mAbs (clone IDEC-131) showed promising results in inhibiting the relapses during clinical trials [9, 18]’.Methodology section was corrected according to all aspects risen by the Reviewer. The sentence: ‘…mixture proteolipids protein…’ has been replaced with the ‘mixture of peptides’ throughout the manuscript. We also used ‘pep(MOG/MBP/PLP)’ as abbreviations. We have removed the paragraphs about ddPCR and NGS analysis. Results: The CFSE proliferation (Figure 1B), % of CD49d+CD154+ lymphocytes (Figure 1A) and sCD40 (Figure 1C) experiments were done independently for each individual in both groups (RR-MS and HC). Therefore, we have performed statistical analysis, and a new graph showing the differences has been added to the Figure 1B. Appropriate changes were done in the figure legends: ‘Bars are presented as means ± SD calculated from ten independent experiments’. Vertical axis for Figure 1A has been corrected. The cytokines were measured by Bio-Plex Pro™ Human Chemokine Assays preciously described in the Materials and Methods (Human Chemokine Multiple Profiling Assays). Additionally information has been added to the Table 1: ‘Cytokines were measured in PBMC supernatants during CD49d+CD154+ lymphocyte expansion with the use of Bio-Plex Pro™ Human Chemokine Assays’
and the Table 2: ‘Cytokines were measured in supernatants of maturating OLs and their co-culture with sorted CD49d+CD154+ lymphocytes by the use of Bio-Plex Pro™ Human Chemokine Assays’
Figure 2B points to the most important findings from the Table 2 to better visualize the shift of cytokines responsible for creating proinflammatory environment by both cell populations. We showed in the Figure 3 that OPCs and CD49d+CD154+ lymphocytes are in close proximity. We also modified the Figure 2B to make it more clear. In the previous version, right panel of Figure 2B contained ‘13 not polarized’ on vertical axis. We have decided to change this graph for hOPC+HC CD49d+CD154+ on the vertical axis vs. hOPC+RR-MS CD49d+CD154+ on the horizontal axis. According to the Reviewer’s suggestion, we have added a new section describing presence of CD49d+CD154+ lymphocytes in the brain (EAE model):3.3. CD49d+CD154+ lymphocytes are present in the EAE mouse brain
‘To confirm whether our in vitro observations are reflected in vivo in brain, we used mouse model of MS (EAE). Firstly, using IHC, we confirmed that leucocytes were present in the EAE brain during remyelination period three weeks after the disease peak (Figure 3A). Secondly, we noted that in EAE, opposite to HC brain, CD49d/CD154 signal was co-localized around dark areas outlined by O4 signal of maturing OPCs (Figure 3B). We also found in EAE that the decrease in O4 signal was accompanied by the increase in CD49d+ and CD154+ signals, while in HCs only O4 signal was detected (Figure 3C left panel). High co-localization of CD49 and CD154 florescence signals in EAE mouse (average overlap coefficient factor >0.93) suggested that signals came from the same double-positive cells. Low co-localization of florescence signals of O4 vs. CD49d+ (average overlap coefficient factor ~0.24) and O4 vs. CD154+ (average overlap coefficient factor <0.15) indicated that the signals came from different cell populations located in close proximity (Figure 3C right panel)’.
We have also shown new data in the Figure 3. We have added new photographs from the microscopy analysis with different magnifications demonstrating inflammatory cell infiltration in EAE brain during disease remission. We have performed statistical analysis of immunofluorescence intensity, and calculated the overlap coefficient factor which additionally confirmed that signal from CD49d and CD154 belongs to the same cells (Figure 3C):‘Figure 3. Presence of CD49d+CD154+ lymphocytes in EAE brain three weeks after the disease peak. (A) Histopathological examination of horizontal brain section demonstrated the presence of inflammatory cells three weeks after EAE peak. (B) IHC microscopy analysis of sagittal brain slices revealed high CD49d (red pseudocolor) and CD154 (green) signals which were co-localized around maturating OPCs (labelled by O4, blue pseudocolor). Green arrows pointed the regions with high double-positive signal for CD49d and CD154 close to the maturating OPCs. (C) O4, CD49d and CD154 fluorescence intensity and overlap coefficient factor analysis. Nonspecific fluorescence (signal noise) was electronically diminished to the level when nonspecific signal was undetectable (background).’
We have checked and corrected language mistakes.Round 2
Reviewer 1 Report
The manuscript has improved. But I still have doubts, mainly with the animal experiments. Please add either data/images underpinning your conclusions or adjust your title/conclusions. Comments:
It was difficult to track your changes upon my previous suggestions on your reply since you did not reply to them in a point-to-point manner. Consider next time doing that to make it easier for the referee to actually see if their raised issues have been adequately addressed.
Missing citation for ARRIVE guidelines.
Fixed in 4% PFA solution is a clumsy term – why not using the most commonly used term formaldehyde solution?
The manuscript has still quite some typos/grammar mistakes.
Figure 3 is improved but is still not too convincing. Why are the brains upside down on panel A? Also, and this is key, you do not show that CD154/49 cells accumulate during remyelination: The mere presence of O4 in your lesion does not say anything about myelin repair, you could simply have OPC or immature OLs which do not myelinate – which makes the title of your article misleading. You would need a marker such as MBP or PLP (or, less preferably, Luxol fast blue or Eriochrome cyanine staining) to say something about the myelin status of the lesion. Also, you just emerge with the concept of overlap coefficient in the results without explaining in the methods how you have done that. And again, your IHC images from panel B are not convincing: O4 should be more intense in white matter.
From your reference: “In general, A2B5+/O4- defines the most immature oligodendrocyte precursors, O4+/O1- defines intermediate precursors, and O1+ defines oligodendrocytes at a more mature stage” – but you did not use the staining in conjunction with O1 so these could also be early oligodendrocytes (which are also O4 positive).
“We agree that the use of nestin or PDGFR labelling would even better show remyelinating areas.” – I did not say that, neither of these markers make a statement about myelin repair because they are markers for OPCs. Remyelination is the repair of myelin by oligodendrocytes after demyelination and you have to show that myelin using a marker for the myelin sheath if you want to say something about remyelination, otherwise your title/conclusion is not appropriate.
Author Response
We have added the citation for ARRIVE guidelines. [21] Kilkenny, C.; Browne, W.; Cuthill, I.C.; Emerson, M.; Altman, D.G. Animal research: Repoting in vivo experiments: The ARRIVE guidelines. Br J Pharmacol. 2010; 160: 1577-1579. We have changed term ‘fixed in 4% PFA solution’ for ‘fixed in 4% formaldehyde solution’. We have checked and corrected language mistakes. The orientation of the brain images in the Figure 3 has been corrected. To demonstrate the presence of CD49d+/CD154+ cells during remyelinization, additional IHC experiments with MBP/CD49d/CD154 labelling (new Figure 4) were done. This experiment confirmed co-localization of CD49d/CD154 signal with high MBP signal. We explained the concept and the method of calculation of overlap coefficient. Relevant information has been added to the paragraph ‘Mouse CNS histopathological examination’: ‘IHC data were additionally presented as the values of overlap coefficient that indicates the overlap of the fluorescence signals between the channels FITC, AlexaFluor 350 and TRITC. It was calculated as the mean value from every single ROI using Leica Microsystem (LAS - X, ver. 3.7.020979 software, Leica, Germany). The overlap coefficient ranges from 0 (no co-localization) to 1 (complete co-localization).’ Overlap coefficient was additionally visualized by cytofluorograms of the signals: CD154 vs. MBP, CD49d vs. CD154, CD49d vs. MBP in EAE and HC brains (new Figure 4C). We also perform histopathology to analyze brain tissues based on the myelin and neuron chemical staining in EAE and HC (new Figure 3A). O4 labelling allowed to demonstrate co-localization of oligodendrocytes progenitors with CD49d+/CD154+ cells. IHC technic we used does not provide a possibility to introduce the 4th color.
Reviewer 3 Report
Dear authors.
The article is much improved. It remains required that you make your experimental procedures more clear to the reader (for every figure and table). In this specifically I mean that you need to state the number of samples used (both control and MS) and if the samples were pooled or tested on individual patients / controls. This will give the reader the context to understand how you derived your statistics and how representative your findings are. The same goes for the EAE study. It is my impression that this is N=1 for a sick mouse and N=1 for a control mouse. If I am incorrect, please make this clear and state your findings clearly. Please also indicate the EAE clinical score(s) for the mouse/mice tested, both at peak disease, and three weeks later at time of remission.
Thank you.
Author Response
According to the Reviewer’s suggestion, we have added more detailed information about experimental conditions. We provided the number of samples used in each experiment in the Figure and Table legends. We also added information about EAE mouse model: clinical scores to the Materials and Methods, and Results; and number of animals used each experiment to Figure 3 and 4 legends.We have checked and corrected language mistakes.
